# Influence of Soil-Borne Inoculum of *Plasmodiophora brassicae* Measured by qPCR on Disease Severity of Clubroot-Resistant Cultivars of Winter Oilseed Rape (*Brassica napus* L.)

**DOI:** 10.3390/pathogens10040433

**Published:** 2021-04-06

**Authors:** Ann-Charlotte Wallenhammar, Zahra Saad Omer, Eva Edin, Anders Jonsson

**Affiliations:** 1Rural Economy and Agricultural Society, HS Konsult AB, Gamla vägen 5G, SE-702 22 Örebro, Sweden; 2Rural Economy and Agricultural Society, HS Konsult AB, P.O. Box 412, SE-751 06 Uppsala, Sweden; zahra.omer@hushallningssallskapet.se; 3Rural Economy and Agricultural Society, HS Konsult AB, Brunnby Gård, SE-725 97 Västerås, Sweden; eva.edin@hushallningssallskapet.se; 4RISE, Research Institutes of Sweden AB, Box 187, SE-532 32 Skara, Sweden; anders.jonsson@ri.se

**Keywords:** *Brassica napus*, clubroot, real-time qPCR, soil-borne inoculum, resistant cultivars, field trials, yield, bioassays, disease severity index (DSI)

## Abstract

Use of resistant cultivars is considered the most effective tool in managing clubroot. Three clubroot-resistant commercial winter oilseed rape (OSR) cultivars and a susceptible ‘Cultivar mix’ were evaluated for disease severity index (DSI) and yield performance in field soils, selected for varying abundance of natural inoculum of *Plasmodiophora brassicae*. Seven field trials were carried out during 2017–2019 in winter OSR crops, and comparative bioassays were performed in a growth chamber. Substantial variation in clubroot infection between years was observed in the field trials. For Cultivar mix, a negative correlation (y = −252.3ln(x) + 58,897.6) was found between inoculum density and seed yield in five trials, whereas no correlation was found for the resistant cultivars. In bioassays, Cultivar mix exhibited a significantly high correlation between DSI_b_ and number of gene copies g^−1^ soil (R^2^ = 0.72). For resistant cvs., Mentor and Alister, correlation was R^2^ = 0.45 and 0.58, respectively, indicating that resistance was under pressure. In field trials, DSI_f_ of the resistant cultivars was lower (<27). The recommendation is thus to use clubroot-resistant cultivars of OSR as part of Integrated Pest Management in situations where abundance of *P. brassicae* DNA exceeds 1300 gene copies g^−1^ soil.

## 1. Introduction

Rapeseed is an important cornerstone in Swedish cropping, with production of rapeseed oil and rapeseed meal providing high-value food and feed constituents with good nutritional composition [1]. Brassica oilseed crops have been important in Swedish farming for the past 80 years [2] and winter oilseed rape (OSR) (*Brassica napus*) is currently a profitable crop, with sharply increased acreage in recent years following considerable damage by insect pests in spring OSR. In the main Swedish production areas, the acreage of winter OSR has increased from 90 to 94% of total oilseed rape acreage since 2015. However, increasing outbreaks of plant diseases that survive in the soil as resting spores or resting bodies, such as clubroot, Sclerotinia stem rot, and Verticillium wilt, have also been reported recently in Sweden and globally [3].

Clubroot is the most serious soil-borne disease of brassica oilseed and vegetable crops world-wide, causing appreciable yield losses [4]. The pathogen is an agricultural and biological challenge [5]. The causal agent, the obligate biotroph protist *Plasmodiophora brassicae*, is a member of the eukaryotic kingdom of Rhiazara, in the novel clade Phytorhiza [6]. The 25.5 Mb genome sequence of *P. brassicae* single spore isolate e3 was presented for the first time a few years ago [7]. Internationally, the spread of *P. brassicae* has increased rapidly in spring OSR (canola) in Canada [8,9], North Dakota [10], Finland [11] and Latin-America [12], and in winter OSR in the UK [13], Germany [14], Poland, [15,16], Czech Republic [17], China [18], and other countries.

In recent years, disease outbreaks in winter OSR have been frequently observed in commercial crops in Sweden, particularly in south and central parts of the country. These outbreaks, together with results from soil analysis in Scania (Sweden) [19] and in fields throughout the winter OSR area, show that the spread of *P. brassicae* has increased sharply. In DNA analyses by the Swedish Seed and Oilseed Growers’ Association on 210 farm samples collected in 2013–2015, *P. brassicae* DNA was detected in 49% of the fields sampled, but with large variation between fields and regions [20]. Recurring infections are now being recorded in areas in central Sweden where the prevalence of clubroot was high in the 1990s [2].

Control of clubroot is particularly difficult due to the long persistence of the pathogen in the soil [21], which impedes disease control by means of crop rotation. Breeding brassica crops for resistance has been the ultimate objective for many years, but progress is limited [22]. In Sweden, breeding for clubroot resistance in brassica oilseed crops started at Svalöf AB in the 1960s [23] and yielded one resistant cultivar, ‘Tosca’. However, high yield penalty resulted in its withdrawal when the clubroot-resistant cultivar cv. Mendel, with low yield penalty, was released in 2006 by NPZ in Germany. The resistance is based on the dominant gene and was initially effective against most field populations of *P. brassicae* [14]. However, as a consequence of frequent cropping of resistant cultivars in infested soil, varietal resistance is under pressure in some areas of the UK and has broken down in some cases [13]. Pathotypes capable of causing high levels of disease in resistant cultivars were reported in 2013 in the state of Alberta, Canada [24], with isolates found in these fields showing virulence to all resistant cultivars tested. Confirmed cases of virulent isolates infecting cv. Mendel are reported regularly, with most of the virulent isolates originating from North-east Germany [14]. A study in 2015 showed widespread occurrence in Germany of isolates of *P. brassicae* that can infect the resistant cultivar cv. Mendel [25].

Managing *P. brassicae* is one of the major challenges in rapeseed production and active measures must be taken, as the resting spores persist in the soil for up to 17 years [21], with an estimated half-life of 3.6 years. Spread of infected soil by machinery, wind [26,27] and water [28] has been demonstrated. The classic disease symptoms consist of an enlarged root that contains tens of millions of spores per gram of root. The resting spores are robust, with a well-developed mechanism involving various spore walls that provide protection against degradation by soil microorganisms. At high soil moisture and favorable temperature, the resting spores germinate, and zoospores are produced. These infect the root hairs and an intracellular primary phase takes place, followed by release of secondary zoospores [29]. The second phase of infection takes place in rhizodermal cells of the root and the pathogen subsequently induces local hypertrophy caused by mis-regulated plant-derived hormonal pathways [30]. The main part of the life cycle takes place well protected inside the root.

A DNA-based qPCR assay to analyze the presence of *P. brassicae* DNA in soil samples has been developed within BioSoM, a collaborative thematic project run by the Swedish University of Agricultural Sciences and the agricultural industry to devise biological mapping methods for soil-borne pathogens [31]. Using this assay, the abundance of *P. brassicae* DNA in soil samples can be determined at two commercial laboratories in Sweden, Eurofins Food and Testing Sweden AB (www.eurofins.se, accessed on 15 January 2021) and Intertek Scanbi Diagnostics AB (www.scanbidiagnostics.com, accessed on 15 January 2021). This fast and reliable qPCR technique has provided insights into the spread of the pathogen across Sweden, with soil analyses from around 30 variety trials in winter OSR and spring OSR showing contamination in 15% of fields tested [32]. Alarming spread of the disease has been demonstrated, with *P. brassicae* found in 60% of 45 fields on 18 farms in South-west Scania sampled in 2013 [19].

Breeding for resistant cultivars has expanded world-wide as clubroot disease has rapidly spread in oilseed rape production areas world-wide. Since 2012, several resistant winter OSR cultivars, developed based on the cv. Mendel resistance [33], have been available to Swedish OSR growers. However, these resistant cultivars are only partially resistant, designated “varietal resistance” in UK [13], as they are infected to some extent by even low levels of pathogen inoculum [34]. In Swedish winter OSR field trials evaluating clubroot-resistant cultivars harvested in 2014, resistant cultivars showed high yields but also substantial disease incidence, with 16–37% of resistant plants categorized as infected [35]. The highest disease incidence was found at a field trial site with very high abundance of *P. brassicae* DNA, corresponding to 10 million spores per g of soil, which implies considerable propagation of the pathogen in soil following a clubroot-resistant crop. There is a risk of overcoming this resistance [36], as has already occurred in fields with short OSR rotations in Canada.

Results from earlier Swedish studies on the agronomic performance of resistant breeding lines of spring oilseed turnip rape (*Brassica rapa*) are currently used for interpreting the results of soil DNA analysis and for providing guidance to growers. These studies showed that, at extremely high levels of inoculum in the soil, severe infection can occur in resistant breeding lines [34].

In the present study, performance of the resistance trait in commercial clubroot-resistant cultivars of winter OSR was evaluated in field soils with a natural inoculum of *P. brassicae*, to support choice of cultivar with respect to abundance of soil-borne inoculum. The aim was to develop an approach supported by DNA technology for integrated pest management (IPM) of clubroot in winter OSR crops. This was done by assessing infection levels and determining disease severity and yield of resistant and susceptible cultivars of winter OSR grown at field sites with differing abundance of *P. brassicae* DNA, and by performing comparative studies in a growth chamber in a controlled environment, in order to identify the prerequisites for maintaining sustainable production of winter OSR in fields where *P. brassicae* occurs. The hypothesis was that the selection pressure for new virulent pathotypes is considerably higher if resistant cultivars are grown in soils with confirmed abundance of *P. brassicae* DNA.

## 2. Results

### 2.1. Field Trials 2017–2018

#### 2.1.1. Influence of Soil-Borne Inoculum of *P. brassicae* on Agronomic Performance of Winter OSR Cultivars

Four field trials were carried out in 2017. Two fields, located at Simrishamn and Tomelilla in southern Sweden (Scania), were selected for high abundance of *P. brassicae* DNA, with 1,100,000 and 2,500,000 gene copies g^−1^ soil, respectively. A field located at Kumla in central Sweden was selected for moderate abundance of disease inoculum, 15,000 gene copies per g^−1^ soil, and a field at Hallsberg in central Sweden for low levels of disease inoculum, 2500 gene copies g soil^−1^ (see Material and Methods). The three clubroot-resistant cultivars tested performed best at the two sites with the highest inoculum levels (Simrishamn, Tomelilla), with significantly higher seed yield (kg ha^−1^) than a susceptible Cultivar mix (*p* < 0.001) (Appendix A). At the field trial site at Kumla with moderate inoculum level, statistically significant differences in seed yield were only obtained between Cultivar mix and clubroot-resistant cv. Alister (*p* = 0.017). A yield penalty was observed for all three resistant cultivars at the field site with the lowest inoculum level (Hallsberg), where all resistant cultivars produced significantly (*p* < 0.001) lower seed yield than the susceptible Cultivar mix (Appendix A). Seed yield of Cultivar mix at all sites was correlated with the amount of gene copies of *P. brassicae* per g^−1^ soil, with seed yield decreasing rapidly when the number of gene copies g^−1^ soil increased above 4000 (y = −252.3ln(x) + 5897.6; R^2^ = 0.59) (Figure 1a). No significant correlations were observed for the resistant cultivars (Figure 1b). In terms of oil yield, all three clubroot-resistant cultivars had significantly higher oil yield (kg DM ha^−1^) than Cultivar mix at Tomelilla (*p* < 0.001), while at Kumla only cv. Alister differed significantly from Cultivar mix (*p* = 0.011) (Appendix A).

Thousand-seed weight at Kumla was on average 5.7 g for Cultivar mix, 6.4 g for cv. Mentor, 5.7 g for cv. Alister, and 6.5 g for cv. Archimedes. Chlorophyll content was on average 11.2 ppm for Cultivar mix, 13.3 ppm for cv. Mentor, 11.3 ppm for cv. Alister and 14.4 ppm for cv. Archimedes. There was a significant difference (*p* = 0.014) in plant density between autumn count and spring count at the Tomelilla site, with 34% of Cultivar mix plants dying during winter. The winter-kill reduction in number of Cultivar mix plants was 25% at Simrishamn, 22% at Kumla, and 11% at Hallsberg (Appendix A), but there were no significant differences in crop stand between the cultivars in either autumn or spring.

#### 2.1.2. Visual Assessment of Clubroot Disease Severity and Disease Incidence 2017

Disease severity index (DSI) and disease incidence of clubroot varied between the cultivars and were also dependent on location. There was a significant (*p* < 0.001) effect of resistant cultivars on disease incidence and disease severity at the Simrishamn, Tomelilla and Kumla sites, where disease severity index (DSI_f_) of the resistant cultivars was 73–91% lower than that of the susceptible Cultivar mix (Table 1). There was a clear correlation between abundance of soil inoculum as indicated by real-time qPCR (gene copies g^−1^ soil) and DSI_f_ for the susceptible Cultivar mix (y = 5.1678ln(x) − 2.2907; R^2^ = 0.42) (Figure 2a). Disease severity index (DSI_f_) rapidly increased as the number of gene copies g^−1^ soil increased above approximately 4000 gene copies g^−1^ soil. There was no correlation between gene copies g^−1^ soil and DSI_f_ (R^2^ = 0.019) for the resistant cultivars and, even though the roots were infected at a soil inoculum level of 1499 gene copies per g^−1^ soil, DSI_f_ values were below 30 (Figure 2b).

As regards disease incidence, 100% of plants were infected at the Kumla site, although the average number of gene copies determined was 55,600 g^−1^ soil (Table 1). Conducive conditions for disease infection and development prevailed at this field site, sown on 9 August (Appendix A). At the Simrishamn and Tomelilla sites, sowing was delayed due to rain, until 25 August and 26 August, respectively. The reduction in DSI_f_ for resistant cultivars was statistically significant (*p* < 0.001) compared with Cultivar mix (S) and ranged from 77% to 90% at Simrishamn and from 85% to 94% at the Tomelilla site. Disease incidence was high at these sites, with 68% and 63%, respectively, of Cultivar mix plants infected. There were no statistically significant differences in DSI_f_ between the resistant cultivars and Cultivar mix at the Hallsberg site.

### 2.2. Field Trials 2018–2019

#### 2.2.1. Influence of Clubroot on Agronomic Performance of Winter OSR Cultivars

In 2018–2019, three field trials were carried out in fields at Simrishamn and Tomelilla with very high abundance of *P. brassicae* DNA (600,000 and 370,000 gene copies g^−1^ soil, respectively, according to pre-testing) and in a field at Kumla with moderate abundance (50,000 gene copies per g^−1^ soil) (Appendix A). Significantly (*p* < 0.001) lower yield was observed for cv. Archimedes than for the other cultivars at the Kumla site. At the Simrishamn site, significantly higher yield was obtained for Cultivar mix and cv. Mentor than for cv. Alister (R) *(p* < 0.001). There was no significant difference in yield between cultivars at the Tomelilla site. Plant counts to estimate winter damage showed no change in plant density between autumn count and spring count at the Simrishamn and Tomelilla sites, whereas there was a significant difference (*p* < 0.001) in spring count at the Kumla field trial site for cv. Archimedes, with a reduction in plant numbers of 56%. The plant reduction for the other cultivars was on average 17%.

#### 2.2.2. Visual Assessment of Clubroot Severity and Clubroot Incidence

Assessment of disease severity showed no significant difference between cultivars in DSI_f_ in autumn, which ranged from 22.5 for Cultivar mix to 3.3 for cv. Mentor. The assessments indicated that cv. Archimedes suffered from winter damage caused by abiotic factors (Table 2). There was no difference in clubroot severity at the Simrishamn and Tomelilla sites, as DSI_f_ was extremely low (3.3) for Cultivar mix at Simrishamn and no disease was seen at Tomelilla despite the high level of disease inoculum (Table 2). All field trials were assessed for disease a second time, in the stubble after harvest in July 2019, when only 2% and 1% of cv. Mentor plants assessed were infected at Simrishamn and Tomelilla, respectively. Non-conducive conditions prevailed at those sites in 2018 (see Appendix A), as precipitation was low in August and September, while precipitation at the Kumla site after sowing was 39 mm (25–31 August) (Appendix A). All cultivars at Kumla were infected, but no significant difference in DSI_f_ was seen in the autumn assessment, whereas the assessment after harvest showed a statistically significant difference in infection (*p* = 0.016) between Cultivar mix (DSI_f_ 27.5) and cv. Alister (DSI_f_ 1.7) and cv. Archimedes (DSI_f_ 5.3). Average values from both assessments showed significantly (*p* < 0.001) lower DSI_f_ values for the resistant cultivars, 76–85% lower than for the susceptible Cultivar mix (Table 2). Regarding disease incidence, on average 34% of the susceptible plants were infected, while there was significantly lower disease incidence (*p* < 0.004) in the resistant cultivars.

A high correlation between disease severity (DSI_f_) and seed yield was found for the susceptible Cultivar mix (y = −19.683x + 4123.4; R^2^ = 0.66) (Figure 3a). At lower levels of disease severity index (<20), the yield reduction was about 390 kg ha^−1^, while yield was reduced by 2155 kg ha^−1^ (55%) at DSI_f_ close to 100. No correlation was found for the cultivars defined as resistant (Figure 3b). Likewise, there was a significant negative correlation between disease incidence and yield for Cultivar mix (y = −19.013x + 4300.1, R^2^ = 0.66), while no correlation was found for cv. Mentor, cv. Alister, or cv. Archimedes.

### 2.3. Soil Bioassays and Real-Time qPCR Analysis

All cultivars in the field trials were also grown plot-wise in bioassays in a growth chamber under controlled conditions. Real-time qPCR analysis was performed on soil from each plot in the field trials (*n* = 16) (Table 3 and Table 4). In 2017, there was a statistically significant difference (*p* < 0.001) between the field sites in average number (*n* = 4) of gene copies per g^−1^ soil. The number was significantly highest at the Tomelilla site (1,187,100), followed by Simrishamn (419,200), whereas no significant difference was seen between the Kumla and Hallsberg sites (Table 3). No significant differences were found between cultivars (plots), irrespective of field site or number of gene copies per g^−1^ soil. In 2018, a significant difference (*p* < 0.001) between field sites emerged, with the average number (*n* = 4) of gene copies per g^−1^ soil being significantly highest at the Simrishamn site (442,300) (Table 4). There was no significant difference in pathogen abundance between Tomelilla and Kumla soil (38,000 and 3,700 gene copies per g^−1^ soil, respectively). Within the field sites, there was a statistically significant difference in pathogen abundance between samples from plots at Simrishamn *(p* = 0.015) and Tomelilla (*p* < 0.001).

The results of the bioassays in 2017 are displayed in Table 3. Under high disease pressure and conducive conditions, considerably higher disease severity index (DSI_b_) and disease incidence (DI_b_) were seen compared to DSI_f_ for soil samples from Simrishamn and Tomelilla, where DSI_b_ increased by 48% and 36%, respectively, for Cultivar mix and by 270% and 287%, respectively, for the resistant cultivars. In the latter, DSI exceeded 30, which is considered the limit at which resistance is overcome [36]. In soil from the Kumla site, disease severity in the bioassay was 50% lower than that estimated in the field. For Hallsberg samples, disease severity was at a very low level for all cultivars in the bioassay and no statistically significant difference was seen.

The relationship between DSI_b_ and abundance of gene copies in soil for all cultivars tested is shown in Figure 4. A significantly high correlation between DSI_b_ and gene copies was found for Cultivar mix, (R^2^ = 0.7165) (Figure 4a), and for cv. Mentor (R^2^ = 0.4477) (Figure 4b) and cv. Alister (R^2^ = 0.5771) (Figure 4c). The correlation for cv. Archimedes was low (R^2^ = 0.23) (Figure 4d). Considering these cultivars individually at a particular DSI_b_ level, e.g., 20, using the equations in Figure 4a, it was found that it only required 7600 gene copies for Cultivar mix to reach DSI_b_ 20, while it required 1,480,000 gene copies for cv. Mentor, 77,000 gene copies for cv. Alister and approximately 960,000 gene copies for cv. Archimedes. This corresponded to 340,000, 177,000, and 2,200,000 spores per g^−1^ soil, respectively, in the most conducive conditions.

There was a statistically significant correlation between DSI_b_ and number of gene copies for Cultivar mix (*p* = 0.035), cv. Mentor (*p* < 0.001) and cv. Alister (*p* = 0.012) in soil from the Simrishamn site, with a higher number of gene copies in the soil increasing the DSI values. For cv. Archimedes there was no correlation (Table 3). In soil from Kumla, positive correlations were found for cv. Mentor (*p* = 0.011) and cv. Archimedes (*p* = 0.017). There was no positive correlation between DSI_b_ and number of gene copies in soil for any cultivar at any of the sites studied in 2019 (Table 4).

## 3. Discussion

Clubroot is a serious threat to OSR production world-wide. The increasing proportion of clubroot-infected arable land in Sweden has serious consequences for oilseed rape growers [2]. OSR crops are valuable constituents of crop rotations and increasing market prices have resulted in production expanding in recent years, often through increased OSR frequency within rotations in Sweden and other countries [9,13,37,38]. Clubroot-resistant winter OSR cultivars have been available for several years, but the market share is moderate. One constraint is the yield penalty of clubroot-resistant cultivars [35], which was 6–10% relative to susceptible Cultivar mix in the present study when grown in fields with low or no inoculum (Appendix A). In this study, the agronomic performance of clubroot-resistant winter OSR cultivars was assessed in the field on soils with different abundance of *P. brassicae* DNA and in comparative bioassay tests in a growth chamber under conducive conditions for the pathogen.

The severity of clubroot disease varied between the field sites (Table 1), which were selected to represent different levels of disease inoculum determined by real-time qPCR assay [19]. There was also substantial variation in clubroot infection between years (Table 1 and Table 2).

High temperatures in the 2018 season (e.g., 30–37 °C at Kumla) also affected evaporation, and therefore yield differences due to clubroot was probably strongly masked (Appendix A), as experienced by others [3]. Yield in the clubroot-resistant cultivars at Kumla in 2018 was only 10% higher than for the susceptible Cultivar mix, despite severe clubroot infection (Table 1), compared with 35% at Simrishamn and 55% at Tomelilla. Winter OSR sowing date (9 August at Kumla, 25–26 August at Simrishamn and Tomelilla), is likely to have played an important role in the results, as the robustness of surviving plants in spring is crucial for the yield outcome (A. Gunnarsson, Svensk Raps, personal communication 2010). For the susceptible Cultivar mix (cvs. Avatar, Dariot, Explicit, Exstorm) in 2017–2018, a negative correlation (y= −252.3ln(x) + 58,897.6) was found between inoculum abundance and seed yield (Figure 1a). Yield losses caused by clubroot are often not observed in winter OSR [39], but the results in Figure 2 clearly show that at an inoculum abundance of 5000 gene copies g^−1^ soil (corresponding to 12,000 spores g^−1^ soil), estimated yield loss was 8%, while at an abundance of 850,000 gene copies g^−1^ soil (corresponding to 2 million spores g^−1^ soil), estimated yield loss was 44%. For the resistant cultivars, no correlation was found between inoculum abundance and seed yield. Thus, differences in seed yield were due rather to the agronomic performance of the cultivars and the prevailing biotic and abiotic conditions at each site (Figure 1b). The yield reductions observed were in line with those in a field trial on spring OSR (*B. napus*) in central Sweden, where a yield loss of 50% was found at the highest level of disease incidence [39].

There was a clear correlation between amount of soil inoculum (gene copies g^−1^ soil) as indicated by real-time qPCR and DSI_f_ for the susceptible Cultivar mix (Figure 2a). DSI_f_ rapidly increased with increasing pathogen abundance above approximately 4000 gene copies g^−1^ soil (corresponding to 9000 spores g^−1^ soil). There was no such correlation for the resistant cultivars (R^2^ = 0.0192), even though these cultivars were infected at a higher rate, particularly at inoculum densities >70,000 gene copies g^−1^ soil (corresponding to 160,000 spores g^−1^ soil). The DSI_f_ recorded was close to 30 (Figure 2b), indicating that cultivar resistance was under pressure [40] and confirming our initial hypothesis. The clubroot-resistant cultivars showed partial, rather than complete, resistance, as found in earlier studies on summer oilseed turnip rape [34]. The results also support recent findings in a greenhouse set-up that increasing inoculum load applied to the resistant cv. Mendel increases the infection rate [33]. Canadian studies on clubroot-resistant OSR cultivars have found increased inoculum loads (quantified by qPCR) after cultivation and a peak in *P. brassicae* DNA in the following year [41].

Disease development of clubroot is strongly influenced by environmental factors, with soil water content long being recognized as a major abiotic factor contributing to the development of clubroot [42]. Disease symptoms can develop after 10–18 h in excessively wet soil [43]. When soil moisture content exceeds 50% soil water-holding capacity, the disease develops very quickly [4]. Disease assessments at the different field sites in this study clearly showed the influence of soil moisture on infection (Table 1 and Table 2). Despite very high abundance of *P. brassicae* DNA at the Simrishamn and Tomelilla sites in southern Sweden (600,000 and 370,000 gene copies g^−1^ soil, respectively, corresponding to 1.2 million and 740,000 spores g^−1^ soil), disease symptoms were absent in autumn 2018.

The extremely low level of clubroot infection at the sites in southern Sweden in 2018 was associated with exceptionally low soil moisture content after sowing and throughout September (Appendix A). Significantly lower seed yield at Simrishamn was recorded for cv. Alister and cv. Archimedes in 2019, but this was not related to clubroot disease as no infection occurred (Table 2). It was probably due to other soil-borne diseases such as Verticillium wilt, which was prevalent but not assessed. The chlorophyll content was higher at that site (*p* < 0.001), revealing uneven maturity (Appendix A). Seed yield at the Kumla site in 2019 was significantly lower for cv. Archimedes than for all other cultivars. This difference was likely related to winter damage due to vigorous growth in autumn, as plant density in spring was reduced by 56%. Uneven maturity was reflected in significantly higher chlorophyll content (*p* = 0.007).

Temperature is an abiotic factor of great importance for successful clubroot infection. Early studies found that the conducive temperature range for infection is 12–27 °C [44], with galls developing only above 9 °C [44]. More recent studies show that air temperature has a significant effect on clubroot incidence in infected soil, with 40-fold lower risk of clubroot infection at 12 °C compared with 22 °C [11], and with higher clubroot infection at 25 °C compared with 20 °C, particularly at lower pH levels [45].

Winter OSR sown in early August often experiences air temperatures within the conducive range for clubroot infection. In 2017, the field trial site at Kumla was sown when the average air temperature was 20 °C, followed by plentiful precipitation (Appendix A). This resulted in all plants of the susceptible Cultivar mix being infected, with a corresponding DSI_f_ of 92.3.

Recent studies on the survival of *P. brassicae* spores in different temperature regimes show that spores buried in the soil at 4 °C and 20 °C show better survival than spores on the soil surface at −20 °C and 30 °C [46], also ultraviolet (UV) light has been shown to have a negative effect on spore viability.

In the present study, bare fallow preceded the winter OSR crops at the Kumla site in both years, thus exposing the soil surface to UV radiation and absorbing heat. This heat increment may have been detrimental for *P. brassicae* resting spores close to the soil surface, as bioassays and field qPCR measurements showed low levels of *P. brassicae* inoculum (Table 4) relative to number of infected plants in Kumla plots in late autumn (Table 2).

It is evident that, through development of robust resting spores, *P. brassicae* is well-equipped to survive through many seasonal cycles [47] and in hostile environments. The pH at the field sites ranged from 6.1 to 6.9 in topsoil and from 6.8 to 7.4 at 60–90 cm depth (Appendix A), confirming previous observations [48]. However, there was no evident influence of pH on abundance of *P. brassicae* inoculum in soil. Soil water content appeared to be the most important factor for germination, motility of the resting spores and further steps in the infection cycle. During dry seasons, infection was impeded despite a spore load of about 1.4 million spores g soil^−1^ (Table 2). A second assessment after harvest in 2019 (Table 2) confirmed that roots were most vulnerable to infection at the seedling stage [47], as clubroot disease incidence was similar as in late autumn 2018. The infection potential is latent and can lead to a false belief among growers that clubroot is no longer a problem, so soil testing for clubroot infection potential in field soil is crucial for determining choice of cultivar. Use of resistant cultivars is a major tool in IPM [49], but based on findings in the present study these cultivars must be used in environments where infection pressure is low to moderate, as otherwise the resistance is put under pressure (Figure 4b–d). Previous recommendations for Sweden state that an abundance of >325,000 gene copies g^−1^ soil poses a high risk of considerable multiplication of spores in clubroot-resistant cultivars [19]. The results in this study indicate that this threshold should be reduced to 100,000 gene copies g^−1^ soil to avoid a high risk of pathogen multiplication in conducive conditions.

Populations of a new virulence phenotype of the pathogen capable of overcoming resistance in clubroot-resistant cultivars of OSR have been identified in areas where crop rotations are short [13,24]. Recent studies report severe damage from a virulent isolate on clubroot-resistant cv. Mendel at low spore loads [33]. In bioassays in the present study on soil samples from sites where conditions for pathogen infection were optimal, the DSI_b_ values revealed that resistance was under pressure at high inoculum abundance. Thus, virulent isolates infecting resistant cultivars are likely to be prevalent in the Swedish growing districts represented. The physiological specialization of *P. brassicae* at these field trial sites requires urgent investigation.

Under field conditions, DSI_f_ was lower than DSI_b_, but close to the level where resistance of the cultivars tested was under pressure. The influence of high temperatures (<30 °C) on spore survival in the soil and indications of an impact of UV radiation mean that soil sampling to predict the inoculum potential of *P. brassicae* should be undertaken in cool season conditions. Low levels of disease did not seem to affect yield significantly in susceptible cultivars (Figure 3a). Resistant cultivars may achieve high yield despite high inoculum levels, resulting in multiplication of inoculum and putting resistance under pressure. It is therefore critically important for growers to be proactive and use IPM clubroot control tools such as soil tests based on DNA technology and clubroot-resistant cultivars. This approach will help OSR growers in Sweden and elsewhere to overcome the challenges and maintain OSR cultivation in an environment of global warming and climate change.

## 4. Materials and Methods

### 4.1. Field Trial Assessment of Resistant Cultivars of Winter OSR

Four field trials were established in August 2017 and three field trials in August 2018 in winter OSR fields on commercial farms. Sites were selected based on the results of soil testing for prevalence of *P. brassicae* DNA, which was performed at a commercial laboratory, Eurofins Sweden Testing Agri AB, Kristianstad, Sweden (www.eurofins.se, accessed on 14 January 2021). The field sites were in regions with a high risk of clubroot infection and represented a soil content of *P. brassicae* DNA defined as low (<1300 gene copies per g^−1^ soil), moderate (1300–50,000 gene copies per g^−1^ soil), high (50,000–325,000 gene copies g^−1^ soil), and very high (>325,000 gene copies g^−1^ soil), as described in Wallenhammar et al. [19]. Information on the field sites is displayed in Appendix A. Crops were managed by the farmers according to common practices in the region, with application of nitrogen, other nutrients, herbicides, and insecticides. A dose of Fe-III-phosphate (Sluxx^®®^HP, 5 kg ha^−1^) was applied after sowing to prevent snail damage. Fungicide was applied by the field trial managers at growth stage BBCH 65 to ensure that Sclerotinia stem rot did not reduce yield. Growth regulators were not used. The trials were performed in a randomized block design, with four replicates, 24–30 m^2^ plots. The clubroot-resistant winter OSR cultivars cv. Mendel, cv. Alister and cv. Archimedes and a control mixture of four susceptible cultivars (Cultivar mix) were sown annually at a rate of 50 germinable seeds m^2^ and a row spacing of 12 cm. The susceptible Cultivar mix was similar to that used in the Swedish official cultivar trials. In 2017–2018, the cultivars were cv. Avatar, cv. Dariot, cv. Explicit and cv. Exstorm, while in 2018–2019 the cultivars were cv. Avatar, cv. Explicit, cv. Mercedes and cv. Harnas. The seeds were treated with Crusier^®®^ (Thiametoxam 15 mL kg^−1^) in an industrial batch seed treater (Satec Concept^®®^ ML2000, Agritema, Kiev, Ukraine) at the Department of Seed Technology, Rural Economy and Agricultural Society, Bjärred, Sweden. The seed was kindly provided by the seed companies. The plots were sown at the latest by mid-August to ensure that soil temperatures favored infection. The sowing dates in 2017 were 9 August at Kumla, 15 August at Hallsberg, 25 August at Simrishamn and 26 August at Tomelilla. The dates in 2018 were 15 August at Kumla, 17 August at Simrishamn and 21 August at Tomelilla.

Total harvested area for each plot was 18–22 m^2^. The plots were harvested with a plot combine harvester, and stage of maturity, chlorophyll content, yield, oil content and thousand seed weight were determined. Seed purity, water content and oil content were used for calculation of yield (9% wc), oil yield, and oil content (NIT-method ISO 12099) at 9% water content (ISO 12099). Plant density was graded in late autumn and in April the following year. Stage of maturity was graded prior to harvest. Thousand-seed weight and chlorophyll content (spectrometric method ISO 10519:1997) were only measured at Kumla in 2018, but at all field sites in 2019. Meteorological data (temperature and rainfall) for each field site were obtained from Lantmet (www.ffe.slu.se/lm/LMHome.cfm?LMSUB=1, accessed on 16 January 2021). Meteorological conditions during sowing and germination of winter OSR varied between the southern and central regions of Sweden where the field trials were carried out in 2017 and 2018, particularly regarding precipitation (Appendix A).

### 4.2. Soil Sampling and Preparation

#### 4.2.1. Soil Sampling for Selecting Trial Fields

Soil samples to 20 cm depth were collected at 40 randomly selected points along the arms of a “W” sampling pattern, using a soil auger with diameter 22 mm and volume 76 mL (Galko Svets and Lantbruk, Heberg, Sweden). The cores were mixed in a bucket to a composite sample, and handled according to instructions from the laboratory (https://www.svenskraps.se/se/pdf/klumprotsjuka_2020-05-28_Intertek_Analys--Klumprotsjuka_brochure_april%202020.pdf, accessed on 15 January 2021)

#### 4.2.2. Soil Sampling for Determination of Physiochemical Parameters

For determination of physiochemical parameters across the experimental area, 20 soil cores were taken in the top layer (0–30 cm) and 10 cores each in the 30–60 cm and 60–90 cm layers (Appendix A).

#### 4.2.3. Soil Sampling for Bioassays

Immediately prior to sowing, 25–30 soil cores were taken randomly from the top 20 cm of the soil profile and pooled plot-wise for bioassay. The cores were mixed in a bucket into a composite sample of approximately 2 L for the plot and transferred to plastic bags. The soil samples were kept in cold storage until arrival at the laboratory, where the soil was poured into foil tins, clods were macerated by hand and stones were removed prior to drying at room temperature to 5% water content. The dried soil was then stored at 10 °C until the bioassay was carried out.

For the bioassay, the composite sample was poured into a plastic bag closed with a cable tie and placed in an outer plastic bag closed with a bag sealer. The bag was then run in a cement mixer (MEEC 63 l, 220 W, 27.5 rpm) for 5 min and 500 g of the soil were weighed into a 1.0 L plastic container (HPPE-plastic), to which three steel balls (diameter 12 mm) and three screw nuts (14 mm) were added. The container was closed with a lid and shaken in a paint mixing apparatus (Svenska Skandex AB) for 45 s at 1000 rpm. Then 50 g of each mixed sample were transferred to a 50 mL Falcon tube, from which subsamples were used for DNA analysis. The rest of the soil was poured back into the composite sample.

#### 4.2.4. Determination of Soil Chemical Properties

Measurements of selected physiochemical parameters in the soils were performed by Euro Eurofins Agro Testing Sweden AB., Kristianstad, Sweden. Soil pH was determined potentiometrically in water (1.0:2.5, wt/wt).

### 4.3. Soil Bioassay

All cultivars used in the field trials were tested in bioassays. The composite soil sample was divided into four pots (Göttinger, 9 × 9 × 9.5, volume 0.5 L). The bioassays were carried out in a growth chamber in a completely randomized design replicated four times and maintained for a six-week period, to ensure optimal infection according to Wallenhammar [31]. Fifteen seeds per pot of each resistant cultivar or the susceptible Cultivar mix were sown and thinned out to 10 seedlings at true leaf stage. Infected plants were scored according to the following: 0 = no galls; 1 = enlarged lateral roots; 2 = enlarged taproot; 3 = enlarged napiform taproot; 4 = enlarged napiform taproot, lateral roots healthy; 5 = enlarged napiform taproot, lateral roots infected. Disease severity index (DSI) was calculated as
$$\mathrm{DSI}=\;\frac{\sum \;\left(\mathrm{Class}\;\mathrm{no}\right)\;\times \;\left(\mathrm{No}.\;\mathrm{of}\;\mathrm{plants}\;\mathrm{in}\;\mathrm{each}\;\mathrm{class}\right)}{\left(\mathrm{Total}\;\mathrm{no}.\;\mathrm{of}\;\mathrm{plants})\;\times \;(\mathrm{No}.\;\mathrm{classes}\;-\;1\right)}$$

### 4.4. Soil DNA Extraction and Real-Time PCR Analysis

DNA was extracted from 350 mg soil samples (two replicates) using the FastDNA^®®^ spin kit for soil (MP Biomedicals, Solon, OH, USA) according to the manufacturer’s instructions and the modifications reported by Wallenhammar et al. [31]. DNA extracts were eluted in 100 µL DES (DNase/Pyrogen-free water) and further purified once using Wizard DNA Clean-Up System (Promega Corporation, Fitchburg, WI, USA) and three times using Illustra MicroSpin S-300 HR Columns (Cytiva, Marlborough, MA, USA), according to the manufacturers’ instructions.

The DNA samples were analyzed by quantitative real-time PCR (qPCR) using the 7300 Real Time PCR System (Applied Biosystems) and quantified as described by Wallenhammar et al. [31]. DNA from each of the two duplicates was analyzed twice in each qPCR reaction. DNA inhibition was assessed by adding 2 µL of *P. brassicae* standard (150 gene copies) to an extra reaction for one of the duplicate DNA samples and an extra reaction for the negative water control. qPCR analysis performed at the DNA laboratory, Dept. of Soil and Environment, SLU, Skara.

### 4.5. Field Assessment of Disease

An evaluation of disease was performed on 25 plants per plot, which were uprooted in mid-November, placed in plastic bags and brought to the laboratory, where the roots were carefully rinsed in running tap water and assessed for infection. The roots were scored according to the following: 0 = no galls; 1 = slight galls on lateral roots; 2 = moderate galls (<50% of the root system galled); 3 = severe galls (>50% of the root system galled). Field disease severity index (FDSI) was calculated according to Wallenhammar et al. [34], using the equation presented in Section 4.3. In 2019, an additional disease evaluation was undertaken in the stubble after harvest, on 25 plants randomly uprooted and handled as above.

### 4.6. Statistics

The results were processed using ANOVA tests followed by Tukey’s HSD-test (*p* < 0.05) in JMP 15.2 (SAS Inst. Inc. Cary, NC, USA). Interactions between the logarithm of number of gene copies g^−1^ soil and yield, disease severity index and disease incidence for the cultivars in the field trials and bioassays were calculated using regression analysis in JMP 15.2. All data were used in the regression analysis for the bioassays.

## 5. Conclusions

Managing clubroot once it establishes in OSR fields is very difficult due to the persistence of the pathogen in soil. Commercial cultivars with clubroot resistance are available from most seed companies and provide high levels of clubroot control if grown in fields where the pathogen inoculum is low, as determined by soil analysis. The response of resistant cultivars to disease development constitutes the basis for current guidance to growers on the best long-term control strategy for clubroot. New resistant cultivars exhibit higher yield potential, in the same range as susceptible cultivars, and are thus an efficient tool for growers. Soil testing based on DNA technology needs to be used to a greater extent to avoid outbreaks in fields cropped with susceptible cultivars, and consistent loss of valuable seed yield.

## Figures and Tables

**Figure 1 pathogens-10-00433-f001:**
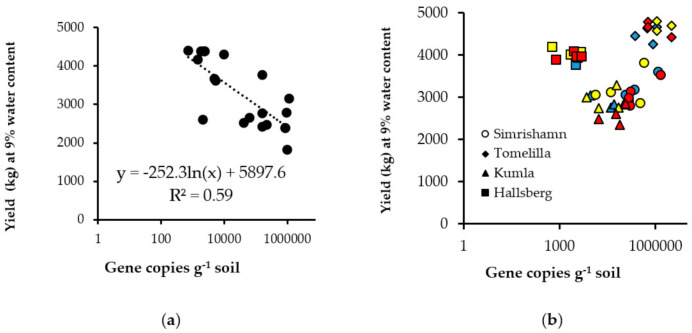
Relationship between gene copies g^−1^ soil (determined by real-time qPCR) and harvested yield (kg, 9% water content) at the four trial sites (Tomelilla, Simrishamn, Kumla and Hallsberg) in 2018 of: (**a**) the susceptible Cultivar mix and (**b**) the clubroot-resistant cultivars cv. Mentor (blue), cv. Alister (yellow) and cv. Archimedes (red). Note: log scale on *x*-axis.

**Figure 2 pathogens-10-00433-f002:**
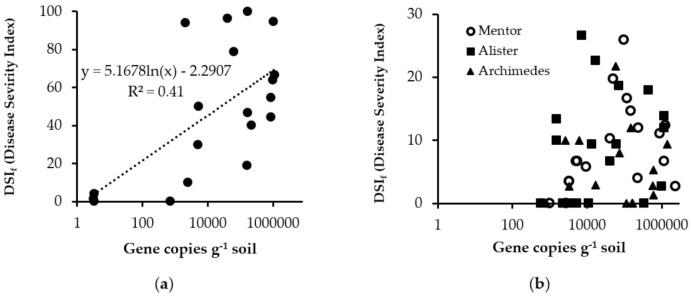
Relationship between gene copies g^−1^ soil (determined by real-time qPCR) and disease severity index in field trials (DSI_f_) for (**a**) the susceptible Cultivar mix and (**b**) the clubroot-resistant cultivars cv. Mentor, cv. Alister, and cv. Archimedes, at all four field trial sites in 2017 and at Kumla in 2018. Note: log scale on *x*-axis.

**Figure 3 pathogens-10-00433-f003:**
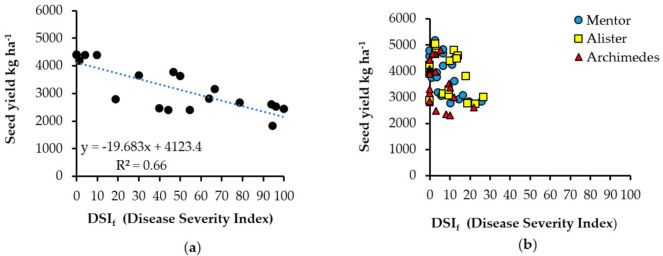
Relationship between disease severity index and seed yield for (**a**) the susceptible Cultivar mix and (**b**) resistant cultivars cv. Mentor, cv. Alister, and cv. Archimedes sown at four different field sites in 2017 and at Kumla in 2018.

**Figure 4 pathogens-10-00433-f004:**
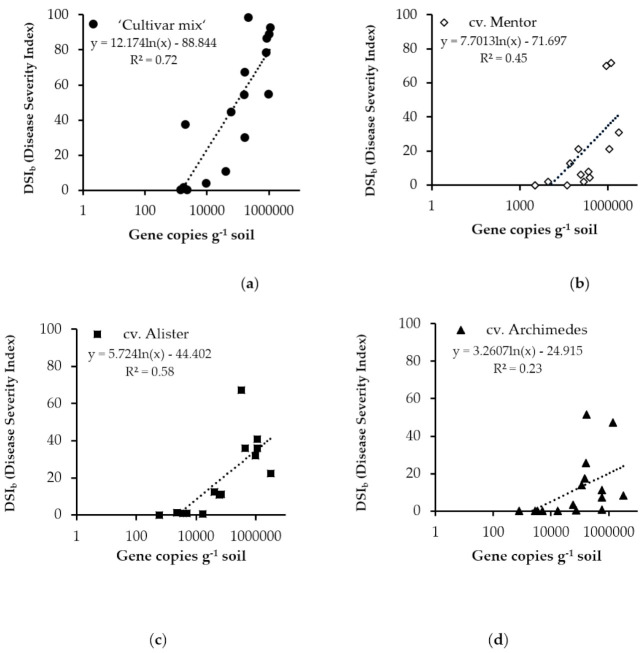
Relationship between number of gene copies g^−1^ soil (determined by real-time qPCR) and disease severity index in bioassay (DSI_b_) in soil from plots with (**a**) susceptible Cultivar mix and clubroot-resistant cultivars (**b**) cv. Mentor, (**c**) cv. Alister and (**d**) cv. Archimedes grown in 2017. Note: log scale on *x*-axis.

**Table 1 pathogens-10-00433-t001:** Disease severity index (DSI_f_) and disease incidence (DI_f_, percentage of diseased plants) of clubroot visually assessed on 15 December 2017 on roots originating from each plot in field trials harvested in 2018 with clubroot-resistant cultivars cv. Mentor, cv. Alister, and cv. Archimedes and susceptible Cultivar mix. Sites were selected based on field abundance (gene copies g^−1^ soil) of *Plasmodiophora brassicae* DNA (Simrishamn 1,100,000; Tomelilla 2,500,000; Kumla 15,000; Hallsberg 2500).

Treatment and Location	DSI_f_		DI_f_ (%)	
*Simrishamn*				
Cultivar mix	52.1	b *	67.9	b
cv. Mentor	12.0	c	15.6	cd
cv. Alister	8.5	c	10.8	cd
cv. Archimedes	5.3	c	6.0	cd
*Tomelilla*				
Cultivar mix	55.4	b	62.5	b
cv. Mentor	8.3	c	11.2	cd
cv. Alister	8.0	c	9.2	cd
cv. Archimedes	2.4	c	4.0	cd
*Kumla*				
Cultivar mix	92.3	a	100.0	a
cv. Mentor	15.5	c	15.1	cd
cv. Alister	19.5	c	16.9	c
cv. Archimedes	8.3	c	6.4	cd
*Hallsberg*				
Cultivar mix	1.8	c	2.7	cd
cv. Mentor	0	c	0	d
cv. Alister	0	c	0	d
cv. Archimedes	0.7	c	2.0	cd
*p-value*	<0.001		<0.001	
Coefficient of variation	56.1		47.4	

* Different letters within each column indicate significant differences according to Tukey’s HSD-test (*p <* 0.05).

**Table 2 pathogens-10-00433-t002:** Disease severity index (DSI_f_) and disease incidence (DI_f_, percentage of diseased plants) of clubroot visually assessed in early December 2018 and July 2019 on winter oilseed rape roots from each plot in field trials 2018/2019 at Simrishamn, Tomelilla, and Kumla for the clubroot-resistant cultivars cv. Mentor, cv. Alister, and cv. Archimedes and the susceptible Cultivar mix. Sites were selected based on field abundance (gene copies g^−1^ soil) of *Plasmodiophora brassicae* DNA (Simrishamn 600,000; Tomelilla 370,000; Kumla 50,000).

	DSI_f_ (0–100)					DI_f_ (%)				
Cultivar	Autumn 2018	July 2019		Both years		Autumn 2018	July 2019		Both years	
*Simrishamn*										
Cultivar mix	3.3	0.0		1.7		5.0	0.0		2.5	
cv. Mentor	0.0	0.7		0.3		0.0	2.0		1.0	
cv. Alister	0.0	0.0		0.0		0.0	0.0		0.0	
cv. Archimedes	0.0	0.0		0.0		0.0	0.0		0.0	
*p-value*	ns	ns		ns		ns	ns		ns	
Coeff. of var. ^†^	400	231		475		400	231		418	
*Tomelilla*										
Cultivar mix	0.0	0.0		0.0		0.0	0.0		0.0	
cv. Mentor	0.0	1.0		0.5		0.0	1.0		1.5	
cv. Alister	1.9	0.0		0.9		5.6	0.0		2.8	
cv. Archimedes	0.0	0.0		0.0		0.0	0.0		0.0	
*p-value*	ns	ns		ns		ns	ns		ns	
Coeff. of var.	231	400		311		231	400		311	
*Kumla*										
Cultivar mix	22.5	27.5	a *	25.0	a	25.0	43.2	a	34.1	a
cv. Mentor	3.3	6.9	ab	5.9	b	7.5	10.9	ab	9.2	b
cv. Alister	5.8	1.7	b	5.1	b	7.5	8.0	ab	4.8	b
cv. Archimedes	6.7	5.3	b	3.8	b	6.7	2.0	b	7.4	b
*p-value*	ns	0.016		0.001		ns	0.025		0.004	
Coeff. of var.	128	99		106		131	109		115	

^†^ Coefficient of variation. * Different letters indicate significant differences according to Tukey’s HSD-test (*p* < 0.05).

**Table 3 pathogens-10-00433-t003:** Disease severity index (DSI_b_) and disease incidence (DI_b_, percentage of diseased plants) of clubroot in bioassays performed in a growth chamber with soil from field trials 2017/2018 with the susceptible Cultivar mix and clubroot-resistant cultivars cv. Mentor, cv. Alister, and cv. Archimedes, and number of gene copies (logarithm at calculation) of *Plasmodiophora brassicae*. Values are mean for soil samples from each plot (n = 4). Sites were selected based on field abundance (gene copies g^−1^ soil) of *Plasmodiophora brassicae* DNA (Simrishamn 1,100,000; Tomelilla 2,500,000; Kumla 15,000; Hallsberg 2500).

Treatment	DSI_b_ (0–100)		DSI_b_ (%)		Gene Copies (No. g soil^−1^)		DSI_b_ × Gene Copies	DI_b_ × Gene Copies
Gene copies *P. brassicae*, average:								
Simrishamn					419,200	b *		
Tomelilla					1,187,100	a		
Kumla					55,600	c		
Hallsberg					2800	c		
*p-value*					<0.001			
Coeff. of var. ^†^					119			
Field-wise:								
*Simrishamn*								
Cultivar mix	77.5	a *	94.5	a	562,300		*p* = 0.035	ns
cv. Mentor	21.9	bcd	30.5	bcd	435,700		*p* < 0.001	*p* < 0.001
cv. Alister	38.6	b	49.2	b	204,100		*p* = 0.012	*p* = 0.027
cv. Archimedes	34.4	b	46.6	b	474,700		ns	ns
*p-value*	0.020		0.020		ns			
Coeff. of var.	141		134		103			
*Tomelilla*								
Cultivar mix	75.2	a	86.4	a	784,300		ns	ns
cv. Mentor	31.6	b	40.2	bc	1,129,800		ns	ns
cv. Alister	32.8	b	42.9	b	1,603,200		ns	ns
cv. Archimedes	7.0	de	16.5	cde	1,231,000		ns	ns
*p-value*	<0.001		ns		ns			
Coeff. of var.	71		66		72			
*Kumla*								
Cultivar mix	29.9	bc	42.0	bc	68,300		ns	ns
cv. Mentor	9.0	cde	14.8	de	49,500		*p* = 0.011	*p* = 0.009
cv. Alister	7.5	cde	9.5	de	38,800		ns	ns
cv. Archimedes	4.4	de	8.2	de	65,800		*p* = 0.017	*p* = 0.009
*p-value*	ns		ns		ns			
Coeff. of var.	323		323		76			
*Hallsberg*								
Cultivar mix	1.3	de	2.8	e	3800		ns	ns
cv. Mentor	0.5	e	1.2	e	1700		ns	ns
cv. Alister	0.8	e	1.8	e	2800		ns	ns
cv. Archimedes	0.0	e	0.0	e	3020		ns	ns
*p-value*	ns		ns		ns			
Coeff. of var.	323		323		77			

^†^ Coefficient of variation. * Different letters indicate significant differences according to Tukey’s HSD-test (*p <* 0.05).

**Table 4 pathogens-10-00433-t004:** Disease severity index (DSI_b_) and disease incidence (DI_b_, percentage of diseased plants) of clubroot in bioassays performed in a growth chamber with soil from the field trials 2018/2019, linked to the number of gene copies (logarithm at calculation) of *Plasmodiophora brassicae* for susceptible Cultivar mix and clubroot-resistant cultivars cv. Mentor, cv. Alister, and cv. Archimedes. Values are mean for all samples from each plot (*n* = 4). Sites were selected based on field abundance (gene copies g^−1^ soil) of *P. brassicae* DNA (Simrishamn 600,000; Tomelilla 370,000; Kumla 50,000).

Treatment	DSI_b_ (0–100)		DI_b_ (%)		Gene Copies (No. g soil^−1^)		DSI_b_ × Gene Copies	DI_b_ × Gene Copies
Gene copies *P. brassicae*, average								
Simrishamn					442,300	a *		
Tomelilla					38,000	b		
Kumla					3700	b		
*P-value*					<0.001			
Coeff. of var. ^†^					113			
Field wise								
*Simrishamn*								
Cultivar mix	15.8	a	19.4	a	633,900	a	ns	ns
cv. Mentor	4.6	a	5.6	b	471,500	ab	ns	ns
cv. Alister	16.2	a	19.2	a	299,900	b	ns	ns
cv. Archimedes	4.0	a	5.8	b	364,300	ab	ns	ns
*p-value*	0.020		0.020		0.015			
Coeff. of var.	141		134		4.0			
*Tomelilla*								
Cultivar mix	54.6	a	61.8		48,400	a	ns	ns
cv. Mentor	22.9	b	25.7		42,400	ab	ns	ns
cv. Alister	29.8	b	33.7		27,300	c	ns	ns
cv. Archimedes	14.6	b	16.9		35,200	bc	ns	ns
*p-value*	<0.001		ns		<0.001			
Coeff. of var.	71.2		66.2		2.9			
*Kumla*								
Cultivar mix	2.5		2.5	a	3400		ns	ns
cv. Mentor	0.7		0.7	ab	3500		ns	ns
cv. Alister	0.0		0.0	b	4000		ns	ns
cv. Archimedes	1.7		1.7	b	3600		ns	ns
*p-value*	ns		ns		ns			
Coeff. of var.	323		323		75			

^†^ Coefficient of variation. * Different letters indicate significant differences according to Tukey’s HSD-test (*p <* 0.05).

## Data Availability

No new data were created or analyzed in this study. Data sharing is not applicable to this article.

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
