# Peer review of "Influence of Soil-Borne Inoculum of Plasmodiophora brassicae Measured by qPCR on Disease Severity of Clubroot-Resistant Cultivars of Winter Oilseed Rape (Brassica napus L.)"

_pathogens, 2021, doi:10.3390/pathogens10040433_

Round 1

Reviewer 1 Report

Dear Authors,

I have reviewed your manuscript and would like to offer some recommendations for improving it. They are not cardinal.
My recommendations:
1. Part 2. Information about meteorological conditions is not a RESULT of your work. It is better to place this information in the "Materials and Methods" section or in the form of a supplemental file.
2. Line 143. The Latin name of the pathogen should be italicized. I ask you to check the entire text for similar errors.
3. Figure 1. Legend text should be justified. Please check it.
4. The text of the "Results" section is full of tables. I recommend that you move most of the tables to the supplemental file.
5. The manuscript contains many pages with extended blank spaces. Fill in these spaces with text.

Author Response

I have reviewed your manuscript and would like to offer some recommendations for improving it. They are not cardinal. My recommendations:

  1. Part 2. Information about meteorological conditions is not a RESULT of your work. It is better to place this information in the "Materials and Methods" section or in the form of a supplemental file.

Reply: The information about meteorological conditions is now moved to the "Materials and Methods" section.

  1. Line 143. The Latin name of the pathogen should be italicized. I ask you to check the entire text for similar errors.

Reply: Thank you for noticing, we have changed that.

  1. Figure 1. Legend text should be justified. Please check it.

Reply: The legend text was checked.

  1. The text of the "Results" section is full of tables. I recommend that you move most of the tables to the supplemental file.

Reply: we have moved table 1, table 3 and table 7 as well as figure 1 to the supplemental file.

  1. The manuscript contains many pages with extended blank spaces. Fill in these spaces with text.

Reply: We have removed the extended blank spaces.

Reviewer 2 Report

This paper describes the qPCR- and disease severety-based assessment of clubroot-resistant cultivars of winter oilseed rape using field trials. Clubroot is a devastating disease of oilseed rape. Although resistant cultivars have been used for the protection of this disease, break down of clubroot resistance often occurs. Therefore, the work is important. The experiments are well-organized and the manuscript is well-written. I have only minor concerns.

Regarding the soil used for field trials, naturally infested soil was used? Or artifically infested? The description on soil is hard to understand which The description on soil is hard to understand which type of infested soil.

L561, p20: The description ‘3 = enlarged napifom taproot; enlarged napiform taproot, lateral root infected’ is confusing and should be re-checked.

Please provide the methods for the measurement of oil content and chlorophyll content.

There are minor typographical errors throughout the text; for example, en dash should be used for page ranges in the references.

Author Response

This paper describes the qPCR- and disease severety-based assessment of clubroot-resistant cultivars of winter oilseed rape using field trials. Clubroot is a devastating disease of oilseed rape. Although resistant cultivars have been used for the protection of this disease, break down of clubroot resistance often occurs. Therefore, the work is important. The experiments are well-organized and the manuscript is well-written. I have only minor concerns.

Regarding the soil used for field trials, naturally infested soil was used? Or artifically infested? The description on soil is hard to understand which The description on soil is hard to understand which type of infested soil.

Reply: the soil used in both field trials and bioassays was naturally infested soil.

L561, p20: The description ‘3 = enlarged napifom taproot; enlarged napiform taproot, lateral root infected’ is confusing and should be re-checked.

Reply: the description was re-checked.

Please provide the methods for the measurement of oil content and chlorophyll content.

Reply: The analyses were conducted by a commercial laboratory at Svenska Cereallaboratoriet AB, Svalöv, Sweden, using NIT-method (ISO 12099) for oil content and spectrometric method (ISO 10519:1997) for chlorophyll content.

There are minor typographical errors throughout the text; for example, en dash should be used for page ranges in the references.

Reply: we have used dash for page ranges in the references.

This manuscript is a resubmission of an earlier submission. The following is a list of the peer review reports and author responses from that submission.